# G Protein-Coupled Receptor 40 Agonist LY2922470 Alleviates Ischemic-Stroke-Induced Acute Brain Injury and Functional Alterations in Mice

**DOI:** 10.3390/ijms241512244

**Published:** 2023-07-31

**Authors:** Yingyu Lu, Wanlu Zhou, Qinghua Cui, Chunmei Cui

**Affiliations:** Department of Biomedical Informatics, State Key Laboratory of Vascular Homeostasis and Remodeling, School of Basic Medical Sciences, Peking University, 38 Xueyuan Rd., Beijing 100191, China; lyydeer@bjmu.edu.cn (Y.L.); wanlu_zhou@163.com (W.Z.)

**Keywords:** stroke, LY2922470, ischemic stroke

## Abstract

Stroke is a major cause of fatalities and disabilities around the world, yet the available treatments for it are still limited. The quest for more efficacious drugs and therapies is still an arduous task. LY2922470 is currently used as a G protein-coupled receptor 40 (GPR40) agonist for the treatment of type 2 diabetes. Previous studies have reported protective effects of other GPR40 activators on the brain; however, it remains unclear whether LY2922470 could be a new stroke therapy and improve the stroke-induced brain damage. Here, we first reveal that the transcriptomic gene signature induced by LY2922470 is highly similar to those induced by some agents being involved in defending from cerebrovascular accidents and transient ischemic attacks, including acetylsalicylic acid, progesterone, estradiol, dipyridamole, and dihydroergotamine. This result thus suggests that LY2922470 could have protective effects against ischemic stroke. As a result, further experiments show that giving the small molecule LY2922470 via oral administration or intraperitoneal injection was seen to have a positive effect on neuroprotection with a reduction in infarct size and an improvement in motor skills in mice. Finally, it was demonstrated that LY2922470 could successfully mitigate the harm to the brain caused by ischemic stroke.

## 1. Introduction

Cerebrovascular diseases are a set of diseases that occur in the blood vessels of the brain and result in damage to the brain [1,2]. Stroke, as a major global issue, is the second leading cause of death, the primary source of disability, and is characterized by a high likelihood of recurrence [3,4]. It is projected that by 2030, the fatality rate caused by strokes will grow to 24.9%, with ischemic strokes accounting for around 80% [5,6,7]. Although researchers have gained deeper insight into the mechanism of ischemic stroke, there is still a paucity of therapeutic interventions. Intravenous thrombolysis with a tissue plasminogen activator (tPA) remains the primary treatment for acute stroke [6]. Nevertheless, only less than 10% of patients are eligible for tPA treatment due to the limitation of a narrow time frame for administering tPA, only 4.5 h, and adverse drug reactions like hemorrhaging [8,9,10]. In addition, the brain is an organ of great importance that requires a significant amount of energy and is highly susceptible to ischemia and hypoxia [11,12]. Although restoring blood flow is the principal method of treating stroke, it can also result in cerebral ischemia/reperfusion (I/R) injury, which is the leading cause of severe disability and mortality in patients. Hence, it is still necessary to seek new medicines for ischemic stroke prevention and treatment, as well as to reduce the damage caused by ischemia/reperfusion.

Several studies have been conducted on GPR40 and its potential for protecting against I/R injury and preserving brain function. Methyl palmitate has been demonstrated to protect the heart from reperfusion injury by activating the GPR40-mediated PI3K/AKT signaling pathway [13]. In addition, GPR40 activation has been demonstrated to modulate the PAK4/CREB/KDM6B pathway in Germinal matrix hemorrhage (GMH) rats, which may lead to decreased neuroinflammation and enhanced neurological function [14]. Agonists of GPR40 have been shown to reduce neuronal damage caused by Alzheimer’s disease GPR40 [15]. LY2922470, a novel small-molecule GPR40 activator, has been demonstrated to be a viable treatment option for type 2 diabetes due to its ability to stimulate insulin production [16,17,18]. However, whether LY2922470 has shown protective effects against ischemic stroke and cerebral I/R injury remains unknown. For the above purpose, in this study we first analyzed the correlation of the LY2922470-induced transcriptomic gene signature with that of small molecules in the Connectivity Map (CMap) database. The result showed that the transcriptomic gene signature induced by LY2922470 is highly similar to those induced by acetylsalicylic acid, progesterone, estradiol, dipyridamole, and dihydroergotamine. Given that these small molecules have protective effects on cerebrovascular accident and/or transient ischemic attack, it is thus inferred that LY2922470 could have protective effects on ischemic stroke, which is highly related to ‘cerebrovascular accidence’ and ‘transient ischemic attack’. With a transient middle cerebral artery occlusion/reperfusion (MCAO/R) creature mouse model, we confirmed that LY2922470 can indeed alleviate cerebral ischemic damage and minimize the infarct size. The result facilitates the development of a promising new remedy for treating cerebrovascular diseases and I/R injury.

## 2. Results

### 2.1. Top Small Molecules Showing Similar Induced Gene Signature with LY2922470

Using the approach described in the Methods and Materials section, we calculated the similarity of gene signatures in CMap with LY2922470. As a result, among the top 100 small molecules, we found that acetylsalicylic acid (No.6), progesterone (No.7), estradiol (No.29), dipyridamole (No.36), and dihydroergotamine (No.78) have protective effects on cerebrovascular accidence and/or transient ischemic attack. Figure 1 shows the top 10 small molecules. These results suggest that LY2922470 could have protective effects on ischemic stroke.

### 2.2. Pre-Treatment with LY2922470 Could Lessen Cerebral Ischemic Injury

In order to assess the efficacy of LY2922470 (the molecular structural formula is shown in Appendix A) in protecting brain tissue, mice were pre-treated with the drug for 5 days prior to surgery (Figure 2a), which was conducted with the standard MCAO/R model. Furthermore, we confirmed that the model worked correctly using mouse laser speckle flow imaging (Figure 2b), and the impact of the medication was evaluated by measuring the size of the acute infarct tissue at 24 h post-operation. The IR group had an infarct size of 45.36% ± 7.33%, whereas the infarct sizes of the mice that were orally given LY2922470 with multiple doses were all decreased to varying degrees. The 40 mg/kg LY2922470 drug group exhibited a substantial reduction in the cerebral ischemic infarct area (20.40% ± 2.43%) (Figure 2c,d). TAK875 is an agonist of GPR40 similar to LY2922470 [19], yet its effects on cerebral ischemic injury in mice when given at a dose of 30 mg/kg were nearly unnoticeable.

### 2.3. Acute Protective Effect of LY2922470 on Brain after Stroke

The MCAO/R model was again used to ascertain whether LY2922470 could offer protection to the mouse brain following ischemic stroke. LY2922470 was injected into the peritoneal cavity at a dosage of 10 mg/kg at 0 h and 12 h post-operation (Figure 3a). The LY2922470 group showed a marked decrease of 52.06% in the size of cerebral ischemia compared to the IR group injected with the control solvent alone (Figure 3b,c).

The only approved thrombolytic drug for the treatment of acute ischemic stroke is tissue plasminogen activator (tPA); however, it has a limited time frame to be effective, as well as coupled with the risk of bleeding [5]. tPA (10 mg/kg) was administered intravenously through the tail 45 min after the restoration of blood flow, with 10% dispensed slowly over one minute and the remaining 90% delivered by a steady pump for 30 min. The combination group is the mice that received both LY2922470 and tPA after surgery. No significant disparity in cerebral infarct magnitude was observed between the three groups (Figure 3d,e).

An examination of the impact of LY2922470 on brain function in mice was conducted using a grip test, which was used to demonstrate functional modifications resulting from a stroke. Although functional impairment presented both with or without LY2922470, the LY2922470 group showed a trend of mitigating the brain function alterations (Appendix A). The data indicated that LY2922470 might guard against the impairment of brain function in mice during the early stages of stroke.

## 3. Methods and Materials

### 3.1. Screening Small Molecules Showing Transcriptomic Gene Signatures Similar to LY2922470

We obtained the LY2922470-induced gene fold change profile (dose: 1 nm, treatment time: 48 h, cell line: A549) from the commercial transcriptome platform from JeaMoon Technology Co., Ltd (Beijing, China). Using the fold change of 1.5 as the threshold, we took the upregulated genes/downregulated genes as the LY2922470-induced gene signature. Next, we performed Fisher’s exact test on the LY2922470-induced gene signature and the small molecules induced gene signatures from CMap [20]. Finally, we evaluated the similarity based on the results of Fisher’s exact tests.

### 3.2. Experimental Animals

The C57BL/6J male mice (Sibeifu Animal Experiment Co., Ltd., Beijing, China) used in this study were 6–8 weeks old. Animals were housed in an SPF animal room, and test mice were placed in individual ventilated cages. During the experiment, the temperature of the animal room was set at 18–26 °C and the humidity was set at 40–70%. Light alternated between light and dark for 12 h. Animals were kept away from radiation during the whole feeding process. Feed water sources for animals were in accordance with the guidelines for the feeding and use of experimental animals. The cages of the experimental animals were replaced once a week. All animal procedures were approved by the Institutional Animal Care and Use Committee of the Institute of Laboratory Animal Science, Peking Union Medical College (IACUC Approval No. YXKT2022L025). All of the experiments complied with all relevant ethical regulations.

### 3.3. Drug Preparation and Treatments

An amount of 160 mg of LY2922470 was added to 800 μL of DMSO solution, and then added to 40 mL of 0.8%CMC-Na solution in batches. After heating and ultrasound for 5 h, a white suspension of 4 mg/mL was prepared. The 0.8% CMC-Na solution was then added to the mixture, resulting in two suspensions with concentrations of 2 mg/mL and 1 mg/mL, and both suspensions were kept stored at 4 °C until needed.

### 3.4. Middle Cerebral Artery Occlusion (MCAO)

A model of middle cerebral artery occlusion and reperfusion was established by temporarily obstructing the left middle cerebral artery. Mice were anesthetized with 2% isoflurane before surgery. The thread (Cinontech, 1220-A5, Beijing, China) was inserted via the external carotid artery into the middle cerebral artery (MCA) and then through the internal carotid artery to create left cerebral ischemia. After 60 or 45 min of embolization, the thread was removed to complete the I/R in the left cerebral area. Regional cerebral blood flow was assessed using a small animal laser speckle flow imaging system (RWD, RFLSI III, China). Sham-operated mice were given the same anesthetic and operation, yet without occlusion of the middle cerebral artery.

### 3.5. Infarct Volume Measurements

TTC staining was used to detect the ischemic area. Animals were given a 10% chloral hydrate solution intraperitoneally, and the brains were rapidly removed after cardiac saline perfusion. The integrity of the brain should be maintained during brain retrieval, and the brain should be rapidly frozen in a refrigerator at −20 °C for 20 min. Six to seven brain slices were cut, one slice at 2 mm intervals. The sections were placed in a 2% tetrazolium bromide (TTC) solution in a water bath at 37 °C in the dark for 10 min, and the container was slightly shaken every 5 min to allow sufficient staining before photographs were taken. Normal brain tissue appeared red, whereas infarcted tissue appeared gray.

After scanning, Image-Pro Plus Image analysis software (https://www.bioimager.com/product/image-pro-image-pro-plus-analyzer-software/, accessed on 5 June 2023) was used to calculate the area of each part. The percentage of infarct size was calculated by comparing the area of the infarct tissue with the normal tissue area.

### 3.6. Statistical Analysis

GraphPad software Prism statistical software (https://www.graphpad.com/features, accessed on 5 June 2023) was used for statistical analysis, and the measurement data of each group were represented by mean ± SEM. ANOVA and *t*-test were separately used for comparison of two groups and more than two groups. A significance level of *p* < 0.05 was considered statistically significant.

## 4. Conclusions and Discussion

Considering the high incidence of stroke and its serious impact on patients’ quality of life, it is essential to discover effective medications to treat stroke and preserve brain function. Our research firstly demonstrated that LY2922470, an activator of GPR40, was able to efficiently diminish the extent of the infarct in mice with acute ischemic stroke, thus providing a possible therapeutic agent for the management of ischemic stroke.

This research focused on a brief evaluation of the pharmacodynamic effects of LY2922470. We administered the drug through both oral and intraperitoneal routes, with distinct doses, frequencies, and treatment durations. The results showed that taking 40 mg/kg of LY2922470 orally and injecting 10 mg/kg of LY2922470 intraperitoneally both yielded remarkable outcomes. Still, further research should be conducted to establish more accurate dosing protocols, including examining the permeability of drugs across the blood-brain barrier and the impact of a stroke on the rate of drug absorption.

Although the results showed a protective effect of LY2922479 against cerebral ischemia, the underlying mechanism of how it alleviates stroke-induced brain damage remains unclear. Exploring the possibility of activating GPR40 is the initial step to take, and a number of other GPR40 activators have been known to possess neuroprotective and cardiovascular protective properties in previous studies [21,22]. The binding of diverse ligands to GPR40 triggers the interaction of various intracellular proteins (like Gα_q_, Gα_s_, and β-arrestin 1/2) with the receptor, which consequently causes the initiation of distinct signaling pathways within the cell [22,23]. However, the GPR40 agonist TAK875 activates only the intracellular Gα_q_ protein-mediated signaling pathways [15]. This “biased agonism” of GPR40 may be a factor in the pronounced disparity in the efficacy of TAK875 and LY2922470 for ischemic stroke. Nevertheless, these sensible inferences are only a component of the hypothesis, and further investigation is required to delve into the particulars of how LY2922470 is effective in treating stroke.

## Figures and Tables

**Figure 1 ijms-24-12244-f001:**
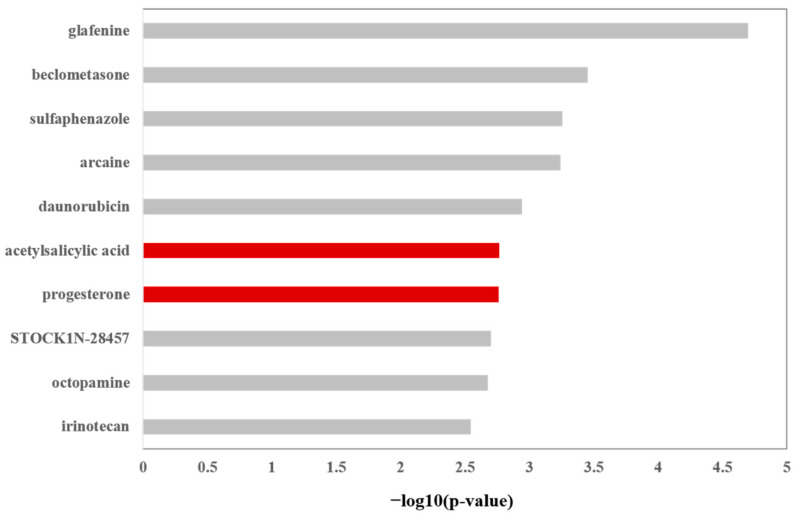
The top 10 small molecules in CMap-induced gene signatures showing highest, similar to those of LY2922470.

**Figure 2 ijms-24-12244-f002:**
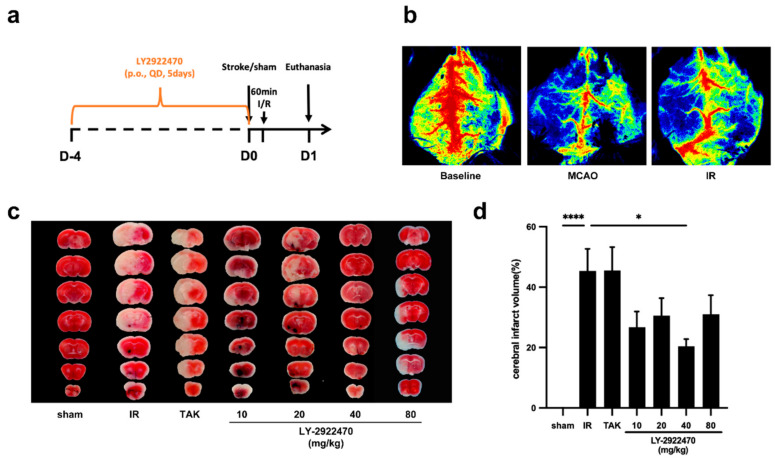
The effect of LY2922470 prevention on acute ischemia stroke. (**a**) Schematic of experimental protocol. Except for the sham-operation group, the mice were treated with TAK875 (30 mg/kg) and LY2922470 (10 mg/kg, 20 mg/kg, 40 mg/kg, and 80 mg/kg). The sham-operation control group and the IR model group were only treated with drug-free solvents (p.o., QD). After the fifth day, the mice were once more administered the drug, and two hours later the MCAO/R mouse cerebral ischemia/reperfusion model was initiated. (**b**) Typical images from laser speckle blood flow imaging. (**c**) Images of brain slices which display the infarct volume at 24 h of reperfusion. (**d**) The quantity of infarction volume in different groups (IR, TAK875 group, *n* = 12; other groups, *n* = 8). Data are mean  ±  SEM from the indicated number of biological replicates: * *p* < 0.05; **** *p* < 0.0001.

**Figure 3 ijms-24-12244-f003:**
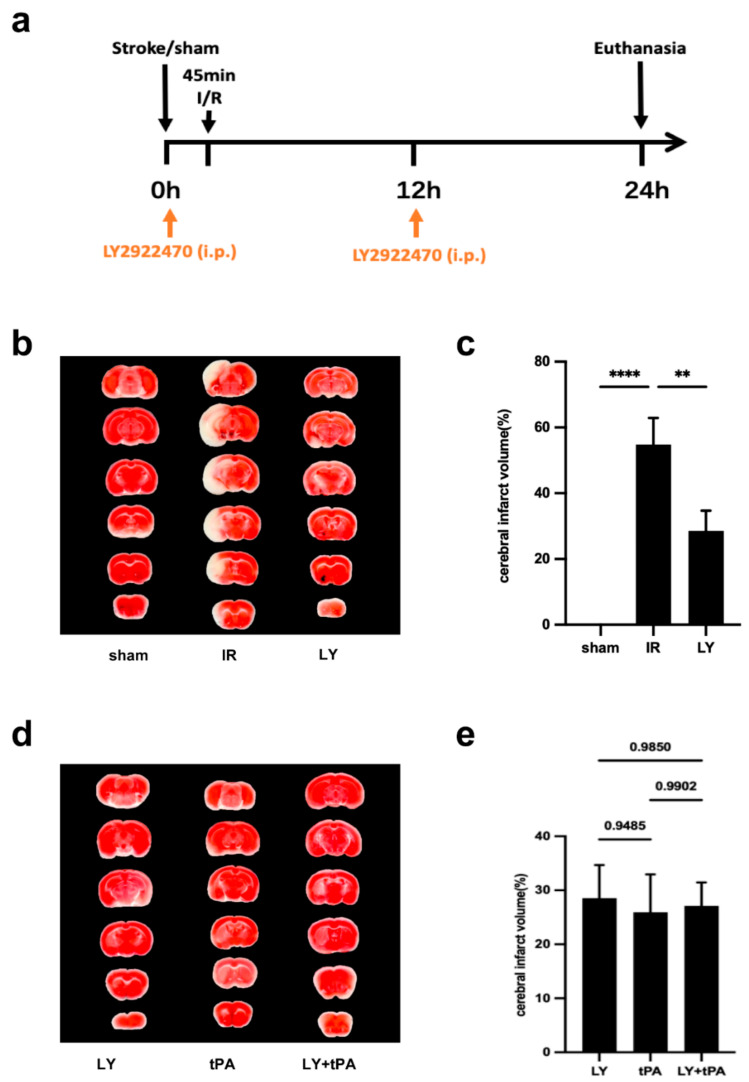
Comparison of the effect of LY2922470 treatment and other medicine on acute stroke. (**a**) Schematic of experimental protocol. (**b**,**d**) Representative photographs of TTC-stained brain sections. (**c**,**e**) The quantification of infarct size in (**b**,**d**), respectively (LY, tPA group, *n*  =  8; other groups, *n* = 7). Data are mean  ±  SEM from the indicated number of biological replicates: ** *p*  <  0.01; **** *p* < 0.0001.

## Data Availability

The data presented in this study are available on request from the corresponding author.

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
