# Peer review of "G Protein-Coupled Receptor 40 Agonist LY2922470 Alleviates Ischemic-Stroke-Induced Acute Brain Injury and Functional Alterations in Mice"

_ijms, 2023, doi:10.3390/ijms241512244_

Round 1

Reviewer 1 Report

The introduction summarizes their findings, which is inappropriate. Overall there is a lack of connection between author’s previous research and this molecule. It seems like they took a shot in the dark. 

Regarding the methods: 

1. Why did the authors perform the search they did. It appears to us they had this molecule and they were trying to justify it’s use so they performed a gene signature search. The question is so what that LY2922470 has similar signature as these other molecules. It doesn’t mean it will be protective. This is backwards to the scienftic method.

2. DMSO can be both caustic and protective. We have moved away from using it for dissolving agents for our mice. How did the authors control for that? Why were the sham animals not given a solution without the agent. 

We appreciate the aim of the authors but this project is hard to understand in terms of aims and methods. 

There are issues throughout. Such as introduction missing the results

Reviewer 2 Report

Manuscript ID: ijms-2463946

Type of manuscript: Communication

Title: GPR40 agonist LY2922470 alleviates ischemic stroke-induced acute brain 

injury and functional alterations in mice

Authors: Yingyu Lu, Wanlu Zhou, Chunmei Cui *, Qinghua Cui *

Submitted to section: Molecular Neurobiology

Comments to the authors:

In the current manuscript, the authors have tried to illustrate the benefits of LY2922470, an agonist of G protein-coupled receptor 40 (GPR40), which has been proven to be a useful treatment option for type 2 diabetes due to its capacity to exhilarate insulin production. However, the authors have shown the alternative role of the LY2922470 compound against stroke and stroke-induced brain damage.

The authors have analysed the transcriptomic gene signature exhibited upon activation of LY2922470, which is highly complementary with other agents that prevent cerebrovascular accidence and transient ischemic attack. This was a significant discovery, and the authors used it to further investigate this molecule and its effectiveness in preventing brain damage during ischemia-ischemic stroke.

The authors have done great analysis, and they could really show that indeed LY2922470 has a significant effect against ischemia-ischemic stroke. LY292470 effectively decreases the amount of infarct in mice with acute ischemic stroke. This research focused on the pharmacodynamic effects of LY2922470 during ischemic stroke. Authors have used other agonists, such as Fasiglifam/Tak-875, but the effect was not as good as LY2922470. To test their hypothesis, the authors have used a model of middle cerebral artery occlusion and reperfusion (MCAO/R), and mice were pretreated with LY2922470 orally and intraperitoneally. In both cases, the results were very significant. The effect of LY2922470 medication on the ischemic stroke model was analysed by quantifying the size of the acute infarct at 24 hours post-operation.

After testing numerous doses, 40 mg/kg of the LY2922470 pharmacological group shows a significant decrease in the cerebral ischemia infarct area when compared to sham and other agonists like TAK-875. The authors did a fantastic job, and further research will undoubtedly be promising as an alternate treatment for ischemia and stroke..

Major comments:

MCAO/R mice were pretreated with TAK875 (30 mg/kg) and LY2922470 (10 mg/kg, 20 mg/kg, 40 mg/kg, and 80 mg/kg). After the fifth day, the mice were once again administered with the drugs, and two hours later, the MCAO/R mouse cerebral ischemia-reperfusion model was initiated.

  • How did the authors come up with the concept of pretreatment with an agonist for five days, followed by administration of the same substance before surgery? Why did they choose 5 days rather than 48 hours when the LY2922470-induced gene fold change profile (dosage: 1 nm, treatment time: 48 hours, cell line: A549) has already been shown?

  • How did the authors obtain with the drug concentrations of 10 mg/kg, 20 mg/kg, 40 mg/kg, and 80 mg/kg for LY2922470 dosage in mice?

    In Figures 2 C and D 10, the infarct size is reduced and again increased to 20mg/kg. Is there any explanation why only 40mg/kg worked and not other concentrations?

  • Finally, in this study, the licenced thromobolytic medication tissue plasminogen activator (tPA) was used with LY2922470 combination treatment to protect the mouse brain from ischemic stroke. However, the concentration of tPA was 10 mg/kg, and the concentration of Ly2922470 was similarly 10 mg/kg. Was there any reason to chose 10mg/kg rather than 40mg?

Minocomments: Please check for some typographical and grammatical mistakes in this manuscript.

Overall, I found the manuscript to be quite intriguing, and the effort to be very promising. I appreciate their efforts in writing such a terrific document and giving a nice outcome.

 Please check for some typographical and grammatical mistakes in this manuscript.
